# Geometry Scaling for Externally Balanced Cascade Deterministic Lateral Displacement Microfluidic Separation of Multi-Size Particles [note 1]

**DOI:** 10.3390/mi15030405

**Published:** 2024-03-17

**Authors:** Heyu Yin, Sylmarie Dávila-Montero, Andrew J. Mason

**Affiliations:** 1Department of Electrical and Computer Engineering, Columbia University, New York, NY 10027, USA; hy2693@columbia.edu; 2Department of Electrical and Computer Engineering, The Citadel College, Charleston, SC 29409, USA; sdavilam@citadel.edu; 3Department of Electrical and Computer Engineering, Michigan State University, East Lansing, MI 48824, USA

**Keywords:** deterministic lateral displacement (DLD), particle separation, externally balanced cascade multi-size separation, I-shaped pillar

## Abstract

To non-invasively monitor personal biological and environmental samples in Internet of Things (IoT)-based wearable microfluidic sensing applications, the particle size could be key to sensing, which emphasizes the need for particle size fractionation. Deterministic lateral displacement (DLD) is a microfluidic structure that has shown great potential for the size fractionation of micro- and nano-sized particles. This paper introduces a new externally balanced multi-section cascade DLD approach with a section-scaling technique aimed at expanding the dynamic range of particle size separation. To analyze the design tradeoffs of this new approach, a robust model that also accounts for practical fabrication limits is presented, enabling designers to visualize compromises between the overall device size and the achievement of various performance goals. Furthermore, results show that a wide variety of size fractionation ranges and size separation resolutions can be achieved by cascading multiple sections of an increasingly smaller gap size and critical separation dimension. Model results based on DLD theoretical equations are first presented, followed by model results that apply the scaling restrictions associated with the second order of effects, including practical fabrication limits, the gap/pillar size ratio, and pillar shape.

## 1. Introduction

Point-of-care (PoC) technology for personal health monitoring and environmental monitoring enables wellness management capabilities, including the early detection of diseases, lifestyle supervision, and the reduction of exposure risks [1]. As Internet of Things (IoT) technologies advance, a growing number of IoT-based biosensing and chemical sensing platforms are being developed, all toward the goal of miniaturized, wearable, and personalized health analysis [2,3]. To support IoT-based personalized sensing platforms, the explosion of micro- and nano-scale technologies has accelerated the need for new methods to analyze sample particles at the micron to nanometer scale. Particles of interest span from natural biological particles to foreign and synthetic particles, and particle analysis applications range from scientific exploration in healthcare and environmental studies to the commercialization of instruments. Particle analysis devices must infer the information of real-world significance from particle parameters including size, shape, concentration, and chemical composition. Because the nature and reactivity of particles often vary significantly with their size [4,5], an increasingly important capability is the separation of particles into size-specific bins, commonly referred to as size fractionation. For example, it is well known that the aerodynamic diameter of foreign particles such as particulate matter (PM) greatly determines their penetration into the human respiratory system and their subsequent related impacts on human health. Due to the critical role PM size plays in health impacts, multi-size particle separation, for example, through microfluidic technologies, is strongly desirable, providing valuable analytical capability across many/all size fractions within real-world PM samples [6]. Especially valuable in PM monitoring are technologies that permit the realization of compact instruments allowing susceptible individuals to regularly monitor the air quality within their everyday microenvironments.

Microfluidic devices are well known for their miniaturization potential and high throughput. IoT-based wearable microfluidic sensing can non-invasively process and analyze biological and environmental components that could affect personal health, enabling smart personalized wellness monitoring [7,8,9]. Several microfluidic separation technologies have been introduced to continuously sort or separate sample particles by size fractions. Active separation technologies, such as dielectrophoresis, electrophoresis, acoustophoresis, immunomagnetic force, or optical force [10,11,12,13,14,15], must incorporate an external force, and thus are complex and not readily miniaturizable. However, passive methods, such as those of cyclone separator, impactor, and deterministic lateral displacement (DLD) devices permit miniaturization because they only rely on the internal forces within specifically designed microchannel structures [14,16,17,18]. DLD has shown great results for the separation of micron-scale particles, such as bacteria and blood components, in centimeter-scale devices [19,20]. Utilizing the asymmetric bifurcation of laminar flow around pillar arrays, DLD was first introduced to perform rapid particle separation (40 s) at a high resolution (10 nanometers). DLD is a passive and label-free technique that is widely used because of its simplicity, predictability, and high separation resolution [20]. Several key advances beyond the basic DLD design have been reported, including the use of I-shaped pillars that induce particle rotation and enhance separation efficiency [21,22]. Additionally, nanoscale DLD arrays of uniform gap sizes ranging from 25 to 235 nm have demonstrated the potential for the on-chip sorting and quantification of nanometer-scale biocolloids [23]. Traditional DLD devices inherently separate particles at only one “critical dimension” related to particle size. However, recent work has shown that DLD can be used to fractionate particles into several different size bins by internally cascading hydrodynamically balanced sections of different geometric parameters [24,25]. These internally balanced DLD devices have different sections that separate at a different critical size by varying pillar parameters, but they must continue to process the larger particles separated in prior sections, and thus the pillar gap, which is a vital parameter in size selection, can never be set to be smaller than the largest particle size. As a result, the range of particle sizes that can be separated is very limited or overall channel lengths become inappropriately long.

Working toward a goal of expanding the dynamic range over which particle sizes can effectively be separated into multiple size bins, this paper introduces a new cascade design, called an externally balanced cascade multi-size gap-scaled DLD. In this new approach, each DLD section separates particles at an increasingly smaller critical dimension and then the separated particles are extracted at the end of that section, permitting the next section’s pillar geometries, including its pillar gap, to be optimized for separating smaller particles. By introducing external components to balance the hydraulic resistance in each section of a multi-section DLD system, this device maintains a laminar flow throughout all sections and achieves a combination of important capabilities that no prior device has reported. Namely it can simultaneously provide a wide dynamic range of size fractionation, separate samples into multiple size bins, and separate particles down to a nanometer in size, all while optimizing the total channel length. Expanding on preliminary results [26], this paper introduces an in-depth description of the multi-section design and more thoroughly defines and evaluates the mathematical model that permits the determination of the available design space and the analysis of interactions between design parameters and device dimensions. Results from this model are presented to help designers better understand the design considerations of externally balanced cascade multi-section DLD separators. The presented approach and model enable a new generation of DLD multi-size separators that push the barriers of dynamic range and minimum particle separation size in many IoT and POC applications.

## 2. DLD Theory

The theoretical operation of DLD separation by particle size is well established and relies on the laminar flow through a periodic array of micrometer-scale obstacles, which are typically pillars within a microfluidic flow channel. As shown in Figure 1a, when each row of pillars is offset by a slight angle or gradient, *θ*, the fluid emerging from a gap, *g*, between two pillars will encounter a pillar in the next row and will bifurcate as it moves around the pillar. Within this fluid flow in the axial or flow direction, the *x* direction, particles that are smaller than a critical separation diameter, *D_c_*, which is determined by pillar geometry, will follow the streamlines and go back to their original lanes after a specific number of rows, *N*. In Figure 1a, this “zigzag mode” can be observed for particles with sizes of *D_p_*_3_ and *D_p_*_4_. In contrast, particles of size *D_p_*_2_, which is larger than the critical size *D_c_*, are bumped off their initial flow path and displaced laterally to follow the pillar gradient. Based on theoretical analysis and experimental verification, the critical separation diameter can be calculated by [21,27]
(1)$$Dc=1.4gN^{-0.48}$$
where *g* is the gap between pillars and *N* is the number of rows needed for particles larger than *Dc* to be shifted by one column from their original position. The pillar gradient is given by
(2)$$\tan\left(\theta \right)=1/N$$

The length of a DLD device can be calculated by
(3)$$L=mND_{x}$$
where $D_{x}$ is the center-to-center distance of the pillars in the flow direction and *m* represents the number of row-shifts chosen to ensure an adequate separation at the output.

## 3. Cascade Multi-Section DLD Separator

### 3.1. Externally Balanced Cascade Multi-Section DLD Approach

Many applications can benefit from separating at multiple size thresholds. Although DLD has generally been utilized to separate particles at a single critical dimension, a few efforts toward multi-size DLD separation have been reported. These prior efforts implement multi-section DLD devices where each section separates at a different critical size by varying only the pillar gradient or geometry parameters, resulting in either the separation of only a small dynamic range of a particle size or a higher dynamic range at the cost of very long devices [24,25,27,28,29,30,31,32,33,34]. Importantly, these devices can separate only a small range of particle sizes because the pillar gap can never be set to be smaller than the largest particles in the sample to avoid clogging, and the pillar gap, $g_{k}$, is the most effective parameter for changing the separation sizes. In contrast, the externally balanced approach introduced in this paper cascades multiple sections in a way that the particles separated by each section will be collected at the end of that section. By removing the largest particles after each section, the following section can optimize its geometries, including the pillar gap, for a smaller size range of particle sizes, without concern that larger particles, which have been removed, could clog the channel. Laminar flow is maintained through all sections by introducing external components to balance the hydraulic resistance in each section of a multi-section DLD system. As results below will demonstrate, by eliminating the largest particles before each subsequent section, the design of each section can be optimized for an increasingly smaller range of particle sizes, which greatly expands the design space of each section and allows for a high dynamic range of separation sizes within a reasonable overall channel length.

Figure 1b illustrates an externally balanced multi-section cascade DLD with scaled design geometries in each section permitting multi-size particle separation with a wide dynamic range (*D_p_max_/D_p_min_*). This illustration shows three cascaded adjacent sections where the critical diameter of particle separation has been gradually decreased in each section due to the gap scaling design. Taking Section 2 as an example, a particle of size *D_p_*_2_, which is larger than *D_c_*_2_, will be displaced and collected from the Section 2 output while all particles with a diameter smaller than *D_c_*_2_ will be conducted into Section 3 for further size fractionation.

### 3.2. Fluidic Mechanisms and Design Rules for the Multi-Section Cascade DLD

Due to the limited understanding of flow fields and the particle dynamics in inertial DLD flows with a Reynolds number well above unity, most current DLD devices focus on a Reynolds number smaller than unity where the fluid inside the microchannel will follow a laminar or even creeping flow model [35]. Although fluid-only predictions are insufficient for explaining experimentally observed critical size behavior, simulated measures agree well with the analytical prediction that a finite size limit on experimentally achievable particle separation often uses a row shift range between 0 and 0.1 (0 ≤ ε ≤ 0.1) from an assumed parabolic velocity profile [35,36]. Similarly, this paper targets applications that work in laminar flow or creeping flow, so the Reynolds number was set as smaller than 1 and the row shift was set within the range (0 ≤ ε ≤ 0.1). Although the full understanding of particle mode behaviors remains elusive, decades of work on DLD have concluded that to design a multi-section cascade DLD device, there are three critical parameters that have to be considered: the hydraulic resistance, gap/pillar ratio, and anisotropic affect.

Firstly, the balance of hydraulic resistance across the lateral direction is critical for all DLD designs. Unbalanced hydraulic resistance would contradict the predictable intrinsic particle movement due to the great lateral pressure induced by the resistance difference across the interface. In addition, an awareness of the hydraulic resistance of each component is vital in determining the applied pressure for the successful operation of the DLD system.

The second figure of merit that is essential in the design is the gap/pillar ratio, which not only affects the pressure drop but also determines the surface area-to-volume ratio for a given overall channel geometry [37]. Moreover, varying the aspect ratio also correlates to hydraulic resistance modification. Additionally, pillar shape not only affects how particles will be displaced, but it will also determine the achievable pillar and gap size range due to fabrication limitations.

Thirdly, the anisotropic effect, which will reduce the in situ critical diameter, has to be carefully considered. The most apparent example of the anisotropic effect is the fluidic resistance of an array of obstacles along the channel wall that will introduce a lateral pressure difference and thus induce the anisotropic effect. Thus, the design of pillars on the boundary must be carefully considered to reduce the anisotropic effect [38,39].

Another parameter that affects the device length or critical size behavior is the *D_x_/D_y_* ratio. A modification to row shift fraction can be used to replace the traditional model if one chooses that *D_y_* is not equal to *D_x_* [40]. However, the relationship between the *D_x_*/*D_y_* ratio with *Dc* size has not been fully determined. In this paper, we assume *D_x_* to be equal to *D_y_*.

Channel depth is another important design parameter. Increasing the channel depth will certainly decrease the array’s flow resistance and reduce clogging. However, deeper channels will limit the pillar and/or gap feature size due to practical fabrication limitations associated with the surface aspect ratio. In this paper, to support the goal of a high dynamic range, channel depth was set to be just higher than the largest particle, *D_p_max_* (*D_p_* is defined as the hydrodynamic diameter of the particle) to avoid particle clogging.

To start a DLD device design, the pillar array’s parameters must be set based on the fluidic mechanisms and fabrication design rule. After choosing the design parameter limits for a specific application, the hydraulic resistance of each section can be calculated. Then, a suitable balancing methodology, such as a long serpentine microchannel design or commercially available valves, can be used to balance the hydraulic resistance between sections. Once the pillar array design parameters are defined, the length of each section and then the whole DLD device can be geometrically calculated. Finally, to compare different design parameter choices, the total device length can be set as a figure of merit to analyze the trade-offs. The design parameters used in this paper are summarized in Table 1.

## 4. Multi-Section Mathematical Model

Since shorter devices not only have less flow resistance but also require less space, to develop a model for our gap-scaled multi-size DLD separation device, the overall goal was defined as minimizing the device length while maintaining a high separation performance at each desired size fraction. As shown in Figure 2, our model assumes that each cascaded section of the device separates at an increasingly smaller critical size, and a section scaling factor (SSF) was established to define the critical size ratio between adjacent sections. Thus, for any section, *k*,
(4)$${Dc}_{k+1}={Dc}_{k}/SSF$$

Notice that, for any total number of sections (NoS), the separator will generate an NoS+1 unique size output. Assuming from Figure 2b that the left output of each section is the input to the next cascade section, the gap in any section needs to be larger than the largest incoming particle to avoid clogging, which is determined by the *Dc* of the prior section. Thus,
(5)$$g_{k+1}={Dc}_{k}\cdot \beta$$
where *β* is a design tolerance variable to avoid particle clogging, which is set to 1.1 in this work. If we further assume that the ratio, *γ*, between the pillar size and the pillar gap is a constant across all cascading sections, then we have
(6)$$w_{k}=g_{k}\cdot \gamma_{k}$$
where the optimum choice of $\gamma_{k}$ will be studied in Section 5. In different applications, $\gamma_{k}$ can be set as >1 to reduce fabrication complexity [37], or as <1 if a migration angle adjustment needs to be considered [41].

Defining *Dc*_0_ as the largest particle size at the input of Section 1
*(Dc*_0_
*= D_p_max_*), (4) and (5) can be reformed into general expressions for any *k^th^* section as
(7)$${Dc}_{k}=\frac{{Dc}_{0}}{{SSF}^{k}}\;,\;g_{k}=\beta \cdot \frac{{Dc}_{0}}{{SSF}^{\left(k-1\right)}}$$

The expression in (7) allows us to define the maximum resolution (smallest separated particle size), *D_p_min_*, for a device with NoS sections as
(8)$${D_{p\_min}=Dc}_{Nos}=\frac{{Dc}_{0}}{{SSF}^{Nos}}$$

Due to the known tradeoff in DLD separation between device length and particle resolution, we can complete the model by expressing the length, *L*, of a multi-section device as the sum of the lengths of all *k^th^* sections using (3) and (6), thus
(9)$$L=\sum_{k=1}^{NoS}\left[m\cdot N_{k}\cdot g_{k}\left(1+\gamma_{k}\right)\right]$$
where *N_k_* is related to *Dc_k_* and *g_k_* by (1).

To explore the design space for multi-size DLD separation with gap-scaled cascade sections, these modeling equations were implemented in MATLAB, and the simulation results across several parameters of interest are presented in Section 5 and Section 6. In these simulations, *m* was set to 1, *D_p_max_* was set to 10 μm, and the other design parameters are defined above.

## 5. Preliminary Analysis to Define the Relationship between Design Parameters and Device Length

The externally balanced cascade multi-size gap-scaled DLD model was first simulated to see relationships and design tradeoffs without any practical fabrication limits. The device length, *L*, and the separation device resolution, *D_p_min_*, were explored to define the optimal range of the total number of sections, *NoS*, and the section scale factor, *SSF*. In this preliminary analysis, because the pillar and gap size has no minimum limit, we set $\gamma_{k}$ to 3 as a starting point, based on our previous study [26].

### 5.1. L vs. NoS Relationship for Different SSFs

Figure 3 shows the relationship between the total length of the multi-size DLD device and the *NoS* for different values of the *SSF* ranging from 1.5 to 3.4 with 0.2 steps. This plot shows that the *L* increases with both the *NoS* and *SSF*, as expected. However, the *L* saturates for an *NoS* larger than ~5–10. This reflects the fact that, due to scaling the gap size down, the length of each subsequent section is smaller and smaller, allowing the total length to reach a saturation point that varies with the *SSF*.

### 5.2. L vs. SSF Relationship for Different NoS

Figure 4 plots the device length as a function of the *SSF* for different values of the *NoS* ranging from 1 to 15. This plot shows an interesting behavior of the *SSF*, with a minimum around an SSF = 1.4 and no increase in the *L* for a *NoS* greater than ~5.

The behaviors from Figure 3 and Figure 4 are further clarified by the 3D plot in Figure 5, which shows the *L* as a function of both the *NoS* and the *SSF*. Here, we can see the valley of the lowest *L* values and observe the tradeoff between the *NoS* and the *SSF*.

### 5.3. The Dp_min vs. NoS Relationship for Different SSFs

An important design parameter not considered in Figure 3 and Figure 4 is the device resolution, *D_p_min_*. Figure 6 plots the *D_p_min_* (in log scale) as a function of the *NoS* for various values of *SFFs* ranging from 1.5 to 3.4. This plot is helpful to determine which values of the *NoS* and the *SSF* can achieve the desired final particle size separation resolution. For example, very small values of *SSF* would be unable to achieve a resolution of less than 0.1 μm. This helps to put some bounds on the useful values of the *NoS* and the *SSF*. To better highlight this, *D_p_min_* values of 0.01 µm, 0.1 µm, and 1 µm (error margin ± 15%) were extracted from data and added to the 3D plot in Figure 5. This allows us to see which values of the *NoS* and the *SSF* can achieve the desired separation resolution.

## 6. Analysis of Secondary Design Considerations

In practical implementations of a DLD device, multiple factors we define as “secondary design considerations” can influence and limit the ideal model defined above. In this section, we analyze several of them, including fabrication limits, gamma variations, and pillar shapes.

### 6.1. Fabrication Limit

Both the pillar size and the gap between pillars will experience limits due to resolutions in the fabrication capability. A DLD device is typically fabricated using deep-RIE with a traditional soft-lithography process [18]. Deep-RIE has feature size limitations depending on the etching depth, which will directly restrict the gap/pillar ratio, $\gamma_{k}$. For example, if we design a 15 µm depth channel, the smallest achievable $\gamma_{k}$ is about 0.2 (*w_k_*/(*w_k_* + *g_k_*) > 0.2). E-Beam lithography could provide a higher resolution (sub-micron) but is not suitable for centimeter-sized devices. For some applications that do not require extremely high resolutions, soft lithography or even 3D printing-based DLD devices with over 10 micro resolutions are used. To demonstrate the impact of fabrication limits on multi-size gap scaling, we define parameters *g_k_fablim_* and *w_k_fablim_* as the minimum feature size for the gap and the pillar size that can be achieved by device fabrication, respectively. For example, if we define the fabrication resolution as *F_R_* for a simple pillar shape design, such as circular, triangle and square shapes, *w_k_fablim_* will be equal to *F_R_*. Thus, the *w_k_fablim_* for circular, triangle and square shapes will be the diameter, side size and width that can be fabricated in different fabrication methods. In contrast, for a specific complicated non-spherical pillar shape that is designed for a specific purpose, the *w_k_fablim_* needs to be designed to be several times larger than the *F_R_*, accordingly. Taking an I-shape as an example, in order to introduce particle rotation efficiently, the dimension of the two protrusions that is set as *F_R_* will determine the smallest pillar size (*w_k_fablim_*) which has to be at least three times larger than *F_R_*. The parameters *g_k_fablim_* and *w_k_fablim_* were added to the simulation model as follows to effectively disable scaling once the pillar and/or gap size reaches this fabrication limit.
$$g_{k}<g_{k\_fablim},\;\mathrm{else},\;g_{k}=g_{k\_fablim}$$
(10)$$w_{k}<w_{k\_fablim},\;\mathrm{else},\;w_{k}=w_{k\_fablim}$$

In our model, the parameters *g_k_fablim_* and *w_k_fablim_* can be chosen by the DLD device designer to match the limitations of a given fabrication facility and process flow. In our fabrication facilities (Lurie Nanofabrication Facility and W.M. Keck Microfabrication Facility), according to the MEMS process we chose, for a circle-shaped pillar design, both the smallest gap size and the smallest circle pillar size were set to 1 µm. Figure 7a plots the simulation results after accounting for fabrication limits and shows that these practical limits significantly change the relationship between the *L*, *NoS,* and *SSF* compared to Figure 3. We no longer see the *L* saturate for large values of the *NoS*, and in fact, the *L* grows to 100 s per meter for a large *NoS* and *SSF*. With the modification in (10), once section scaling reaches the fabrication limit, the gap can no longer be scaled down and, according to (1), the number of rows, *N*, grows increasingly larger with each subsequent section. Because the *L* values in meters are impossibly large, the available design space is significantly constrained. To better show what the model predicts for reasonable values of *L*, Figure 7b shows an example where a 0.01 µm resolution is achieved around *L* = 200 mm (depending on the *NoS*) when we zoom into a specific *D_p_min_* value. Note that *NoS* = 1 defines an exceptional, single-section, device that is not practical for achieving high dynamic range goals, so an amount of *NoS* ≥2 was used in this model.

### 6.2. Gamma Variation ${(\gamma }_{k})$

In Section 4, γ was set to a constant that was used across all sections in order to simplify the relationship among the critical design parameters of a multi-section cascaded DLD device. However, the value of γ strongly influences device dimensions, and in light of the impact of fabrication limits, to reduce the total length of the device, the model was modified to permit the γ value to be varied in each section.

The gamma value for each section *k* that achieves the minimum total device length is defined as γ_k_, the ideal gamma for each section. To investigate the optimal values of γ_k_ for each of the sections, a range of values from 0.3 to 3 was used by the model. For any given device resolution, *D_p_min_*, the fabrication limit parameters, and the NoS of interest, the model returns the length of each section for each value of γ evaluated. For example, for applications such as the size fractionation of PM samples, which range from coarse particles (PM_10_) to ultrafine particles (PM_0_._1_) requiring a 0.01 µm particle separation resolution, if the NoS is set to 10, it would result in 11 size fractions of particles with a size dynamic range of 1000. Figure 8 shows the simulation results for this example case (*D_p_min_* = 0.01 µm and NoS = 10). Figure 8 presents the section lengths in millimeters for different values of γ, with a close-up of Section 1, Section 2, Section 3, Section 4, Section 5, Section 6 and Section 7 shown in the inset. For a goal of the smallest overall device length, the smallest length for each section, *L_k_min_*, can be achieved by selecting the ideal $\gamma_{k}$ value. In the case presented in Figure 8, if *k* > 2, the ideal $\gamma_{k}$ is 1, while if *k ≤ 2*, the ideal $\gamma_{k}$ value will be less than 1.

This analysis was repeated for applications such as blood separation, to analyze a variety of biological particles that require a 1 µm separation resolution while maintaining the 11 size fractions. Figure 9a shows the simulation results for this case (*D_p_min_* = 1 µm and NoS = 10), where the specific section that reaches the fabrication limit is indicated. In Figure 9a, the smallest section length, *L_k_min_*, marked by a star, can be achieved by selecting the $\gamma_{k}$ equal to 1 for a *k > 6* and a $\gamma_{k}$ less than 1 for a *k ≤ 6*. Comparing these results to the example case in Figure 8, we see that changing the device resolution, *D_p_min_*, will change the ideal sectional γ_k_. The ideal sectional γ_k_ was also observed to be a function of the NoS, as shown in Figure 9b where the NoS was decreased to five while keeping the *D_p_min_* at 1 µm.

Because the ideal sectional γ_k_ can vary depending on the number of sections and device resolution, we set our model to look for the shortest device length when the ideal sectional γ_k_ is selected for different values of the NoS and *D_p_min_*. Figure 10a shows the results of the shortest device length by adding the shortest section lengths together as a function of the NoS. In general, the total device length can be dramatically decreased by the proper choice of the NoS. From the inset in Figure 10a, results show that increasing the device resolution, *D_p_min_*, will not necessarily increase the device length significantly, if the ideal sectional γ_k_ is selected for each section separately.

It is worth noting that, in some cases, setting the same γ value for each section will not necessarily extend the device length too much but can significantly simplify the design process. Figure 10b–e present the ratio of device length for different device resolution conditions. As a comparison, the shortest device length using ideal sectional γ_k_ for each section, *L*($\gamma_{k}$), is divided by the shortest device length using the same γ, defined as γ’, for all sections, *L*(*γ’*). The *L*($\gamma_{k}$)/*L*(γ’) ratio can predict if it is worthwhile to choose different γ values for different sections, which will increase the design complexity. For example, as shown in Figure 10d, for the *D_p_min_* = 0.01 µm case, choosing an ideal gamma per section, γ_k_, could reduce the total device length when the NoS is small; however, when the NoS is greater than five, using an ideal gamma per section no longer reduces the total device length but will increase design complexity. In contrast, as shown in Figure 10b, for the *D_p_min_* = 1 µm case, when the NoS is larger than five, utilizing an ideal gamma strategy can save ~25% of the device length. Due to fabrication limits being reached at larger NoS values for the *D_p_min_* = 1 µm case, there is greater variation in optimal gamma values depending on the NoS. Therefore, the *L*($\gamma_{k}$)/*L*(γ’) ratio shown in Figure 10b does not follow a smooth trendline compared to Figure c, d. In summary, as shown in Figure 10e, for small *D_p_min_* cases, where the gap and pillar width will reach the fab limitation faster, very little length reduction is achieved by utilizing the ideal sectional gamma strategy; however, for lower resolution cases, adding more sections with the ideal sectional gamma strategy not only provides more size bins but also achieves an up to ~25% length reduction.

### 6.3. I-Shaped Pillar (More Complex Pillar Geometry)

Different pillar shapes, such as circle shape, triangular shape, T-shape, L-shape, and I-shape, were studied to identify different advantages for specific applications [22,42]. Different pillar shapes require different smallest feature size fabrication capabilities. In our model, $w_{k\_fablim}$ was used to represent the pillar shape and further study how the shape affects the total device length. In real applications, particles will have varied shapes and can be categorized as non-spherical particles. Circle-shaped pillars are the most common in the DLD literature. However, I-shaped pillars can systematically enhance separation efficiency for non-spherical particles. Although, I-shaped pillars require forming a complex groove shape that restricts fabrication capability. Here, the I-shape is analyzed to show how the pillar shape can also be included in the model.

For the I-shaped pillar DLD design, the model was altered to set the smallest pillar size to 6 µm and the smallest pillar gap to 2 µm. Results for the shortest device length using an ideal sectional γ_k_ for each section are shown in Figure 11a. Here it can be observed that a 0.01 μm resolution can be achieved around *L*($\gamma_{k}$) = 1500 mm in an I-shaped design, whereas the circle-shaped design only requires ~100 mm. Thus, in comparison to the circle-shape, the total DLD device length is about 15 times longer for I-shaped pillars due to the smaller fabrication limit when *D_p_min_* is chosen as 0.01 µm. This suggests a 0.01 μm resolution would be very difficult to achieve with I-shaped pillars. However, a 0.1 μm resolution is achievable in ~14 mm using a *NoS* = 5 and an *SSF* around 2.6. In addition, as shown in the Figure 11a inset, the total device length can be reduced dramatically if we sacrifice resolution. Furthermore, Figure 11b shows the *L*($\gamma_{k}$)/*L*(γ’) ratio has similar trends for a I-shape as it did for a circle-shape in Figure 10. Interestingly, as shown in Figure 11b, choosing the ideal sectional $\gamma_{k}$ value for each section in the I-shaped design can always reduce the total device length. However, as the NoS increases, the length reduction gradually decreases.

## 7. Case Study

With refinements to the model presented in Section 5, the mathematical model can be utilized to estimate the total device length for specific study cases. In this section, three case studies, as shown in Table 2, are presented to better illustrate how the final device length is impacted by choosing specific design parameters. The MATLAB Code is provided as Appendix A.

### 7.1. Case 1

The first case is the design of an I-shaped pillar-based DLD device for an application that requires a size separation dynamic range of 1000 and a *D_p_min_* of 0.01 µm using only one section. This design would separate particles having diameters between 0.01 µm and 10 µm into one bin (output) and those less than 0.01 µm into a different output. As calculated by our model, the total device length is around 60 m. Because a 60 m device length is impossible to fabricate, Case 1 demonstrates that it is not practical to achieve both a high separation resolution (0.01 µm) and a very wide dynamic range (1000) in a device with only one section. However, as Case 2 and Case 3 will show, the externally balanced cascade DLD concept introduced in this paper can meet these goals.

### 7.2. Case 2

The second case aims to again achieve a dynamic range of 1000 but now using a multi-section externally balanced cascade DLD design. Assuming a 100 nm fabrication limit (achievable with EB lithography and Stepper lithography) and circle-shaped pillars, our model was applied to estimate the results for two different *D_p_min_* targets. For a *D_p_min_* = 1 µm, as would be common for biological particles, our model calculates that a cascade DLD device with five sections can achieve these goals with a total length of only ~0.3 mm. Similarly, for a *D_p_min_* = 0.01 µm, which would be reasonable for PM fractionation, our model calculates that a cascade DLD device can achieve this with four sections and a total length of ~41 mm, which is suitable for many applications. Overall, this case study demonstrates that, to expand the dynamic range, the externally balanced cascade DLD design provides solutions without increasing the device length to an impractical size.

### 7.3. Case 3

The use of a high-resolution fabrication process increases both the cost and fabrication complexity. For some applications, such as mineral processing, a 1 µm separation resolution is sufficient, and low-resolution fabrication methods, such as 3D-printing or PDMS-based soft-lithography, with 10 µm fabrication limits can be utilized. Because these low-resolution cases can use polymer or soft materials and do not need to be formed in silicon wafers, restrictions to the total device length are relaxed, and lengths up to many centimeters can be tolerated. Here, the designer can simply choose a suitable *NoS* and *SSF* and set the $\gamma_{k}$ to be constant for each section to achieve design goals with a low fabrication complexity. For an example application case requiring particles to be separated into five size bins with a *D_p_min_* = 1 µm and a dynamic range of 100, our model shows that the DLD device can be implemented in a total device length of ~8 mm.

## 8. Conclusions

This paper introduced a new approach for multi-size DLD separation using gap-scaled cascaded sections, and it defined a detailed model of design parameter interactions for this approach. Simulations of the model illuminate informative relationships between design variables that aid in analyzing design tradeoffs. Applying practical fabrication dimension limits to our model significantly impacts the simulation results of the device lengths and narrows down the available design space. Results were also shown for circle and I-shaped pillars, which further highlight the impacts of design choices. The model and results in this paper can enable the achievement of desired separation size resolutions within tolerable device lengths, expediting the development of IoT-based wearable microfluidic sensing platforms.

## Figures and Tables

**Figure 1 micromachines-15-00405-f001:**
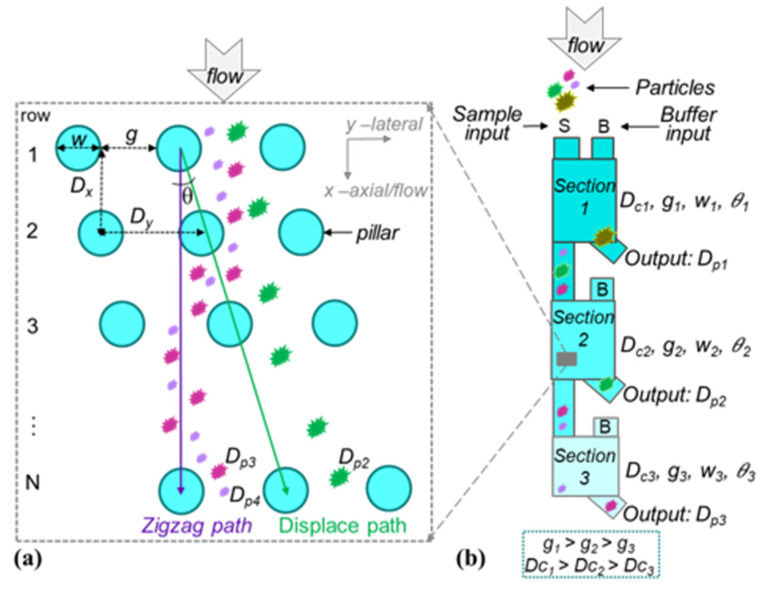
(**a**) An illustration of the pillar array for DLD separation showing key model parameters, and (**b**) a concept diagram for the externally balanced multi-size DLD separator with multiple cascade gap-scaled sections.

**Figure 2 micromachines-15-00405-f002:**
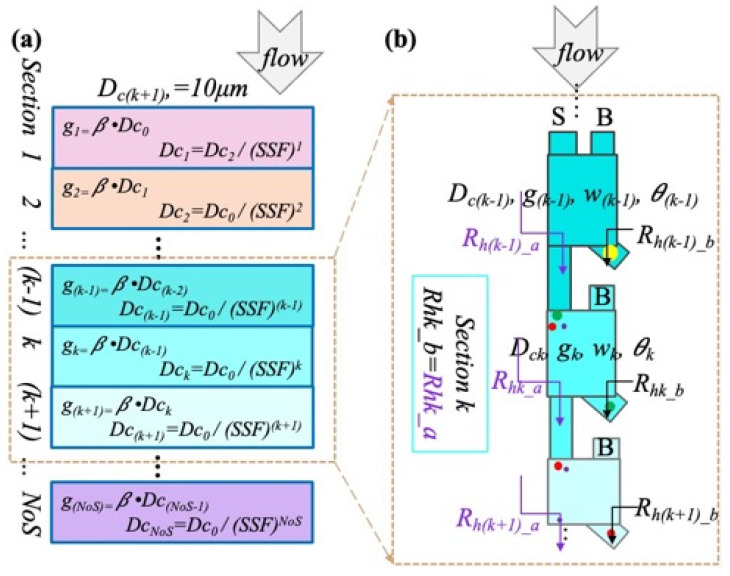
Mathematical model that permits the analysis of design tradeoffs: (**a**) the mathematical model shows the relationship between g and Dc with the SSF and NoS set, and (**b**) section k − 1, k, and k + 1, as examples to illustrate how the hydraulic resistance should be externally balanced.

**Figure 3 micromachines-15-00405-f003:**
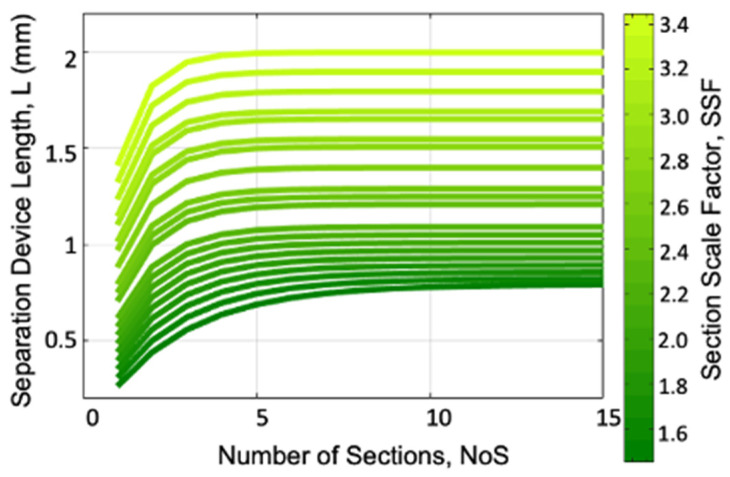
The total device length (L) as a function of the number of sections (NoS) and section scale factor (SSF).

**Figure 4 micromachines-15-00405-f004:**
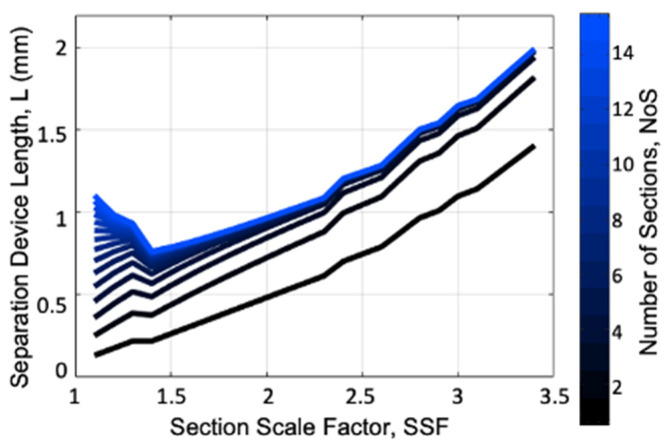
The total device length (L) as a function of the section scale factor (SSF) for different values of the number of sections (NoS).

**Figure 5 micromachines-15-00405-f005:**
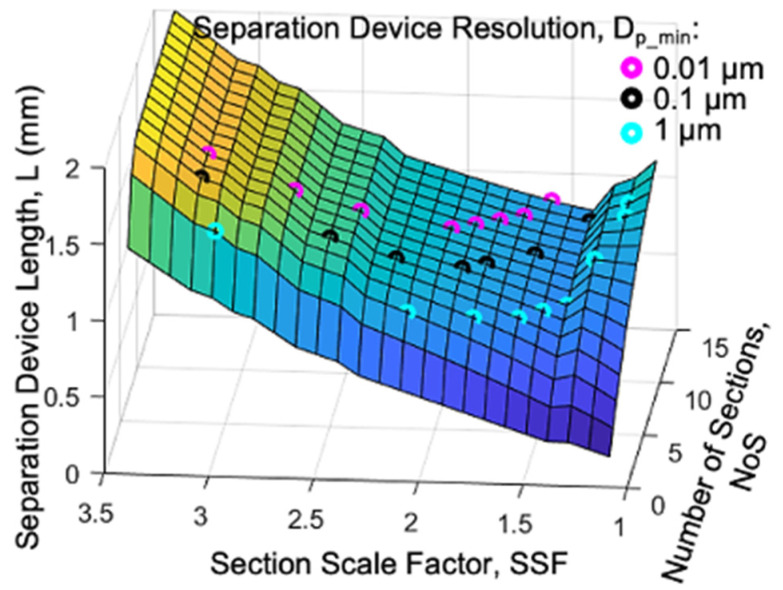
3D plot of the L as a function of both the SSF and the NoS. Colored dots show where various values of device resolution (*D_p_min_*) can be achieved.

**Figure 6 micromachines-15-00405-f006:**
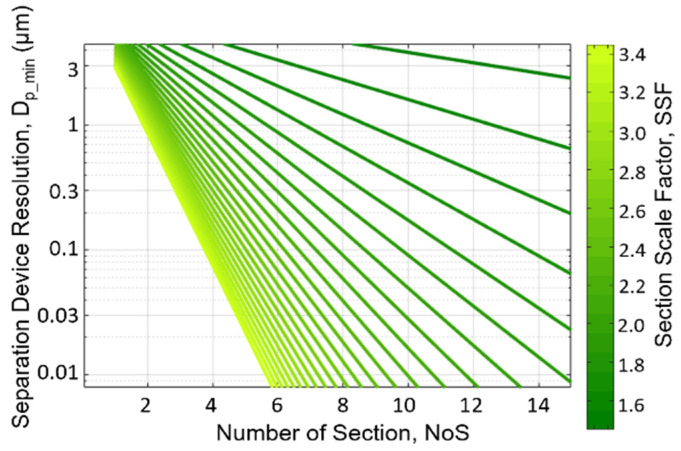
Theory analysis to study how the section numbers (NoS) and section scale factor (SSF) affect the device resolution (*D_p_min_*).

**Figure 7 micromachines-15-00405-f007:**
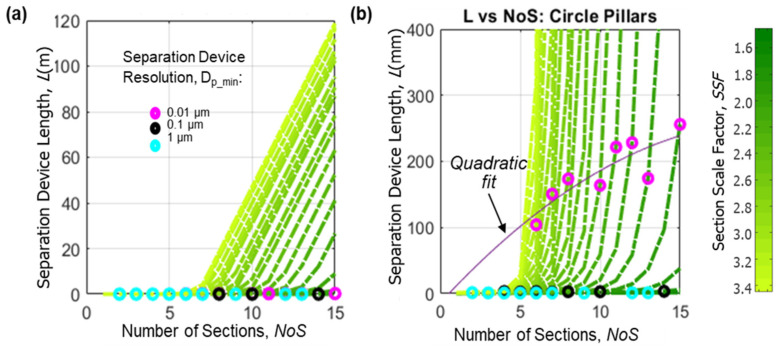
(**a**) The L as a function the *NoS* and *SSF* after implementing practical fabrication limits for circle-shaped pillars. Zooming in (**b**) helps to illustrate the design space suitable for achieving a 10 nm resolution, *D_p_min_*.

**Figure 8 micromachines-15-00405-f008:**
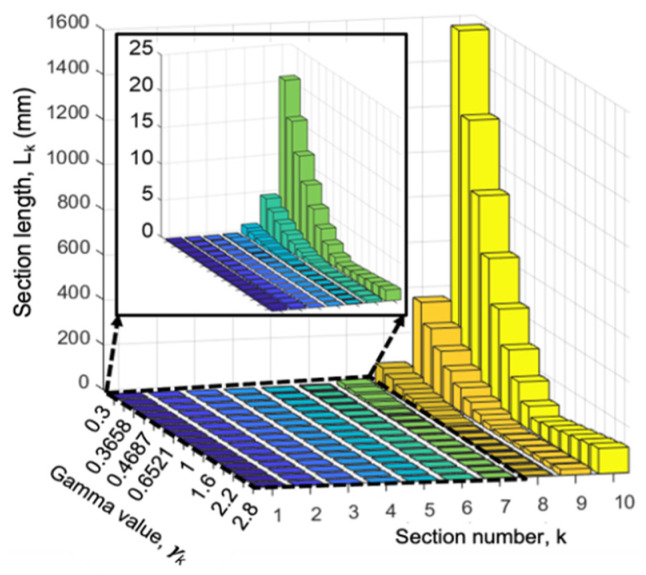
Length of each section, *L_k_*, as a function of $\gamma_{k}$ with implementing practical fabrication limits for circle-shaped pillars when the required *D_p_min_* is 10 nm and the NoS is 10. Details for Section 1, Section 2, Section 3, Section 4, Section 5, Section 6 and Section 7 were zoomed in and presented in the inset.

**Figure 9 micromachines-15-00405-f009:**
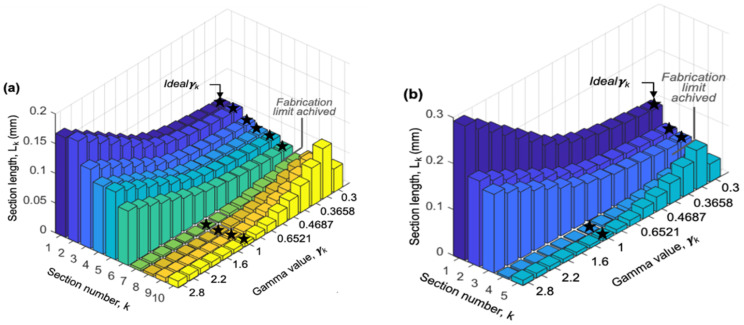
Length of each section, *L_k_*, as a function of $\gamma_{k}$ with implementing practical fabrication limits for circle-shaped pillars when the required *D_p_min_* is 1 µm and the NoS is 10, (**a**), and 5, (**b**).

**Figure 10 micromachines-15-00405-f010:**
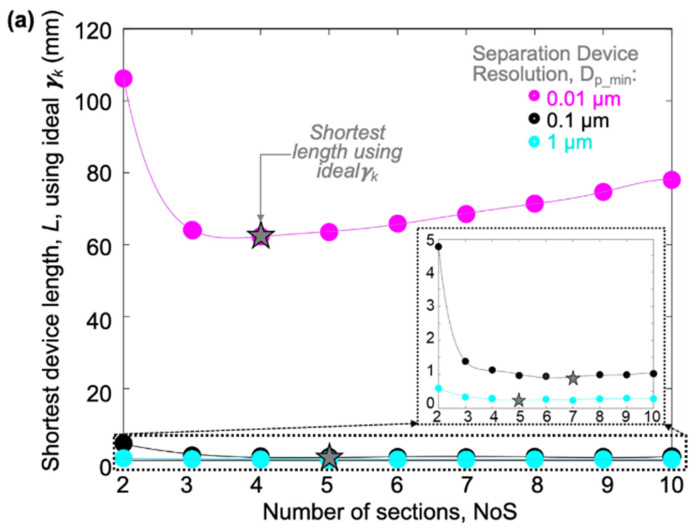
(**a**) Plots the of L vs. NoS for circle-shaped pillars with fabrication limits; different *D_p_min_* values highlight important regions of consideration; (**b**–**e**) Plots of the *L*($\gamma_{k}$)/*L*(*γ’*) ratio vs. the NoS for 1 µm fabrication limit (g_k_fablim_ = w_k_fablim_ = 1 µm) at different device resolutions, where *L*($\gamma_{k}$) is the minimum length using the ideal gamma per section, while *L*(*γ’*) is the minimum length using the same gamma across all sections.

**Figure 11 micromachines-15-00405-f011:**
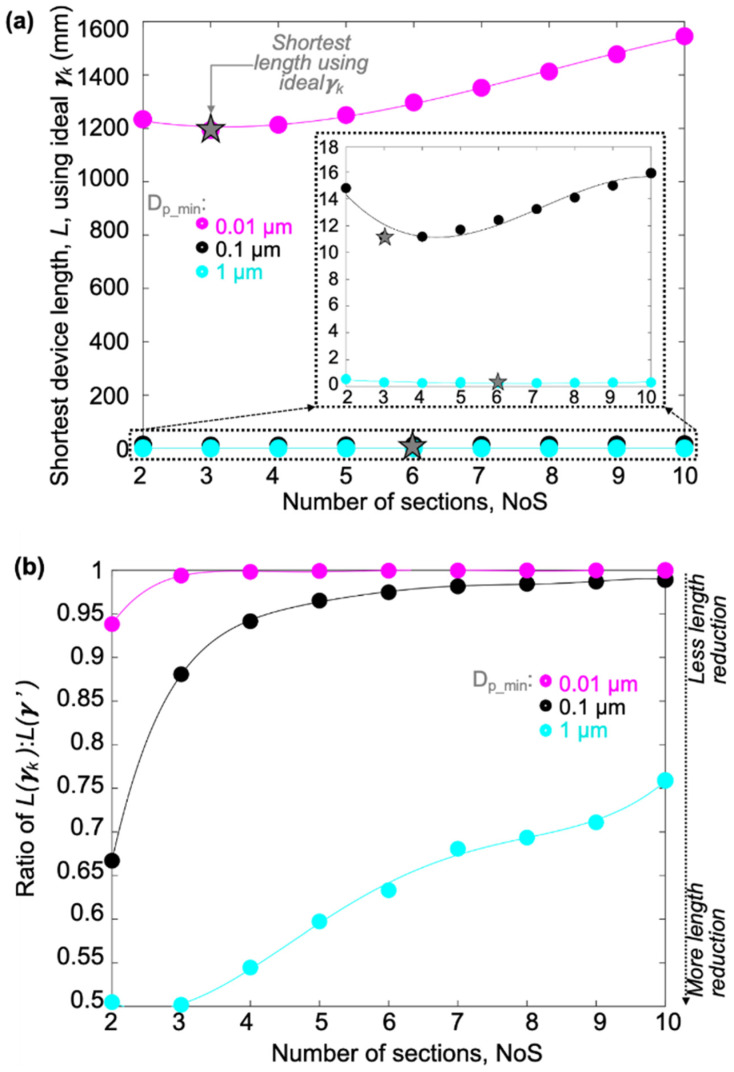
(**a**) Plots of the L vs. NoS for I-shaped pillars with fabrication limits; different *D_p_min_* values highlight important regions of consideration; (**b**) Plots of the *L*($\gamma_{k}$)/*L*(*γ’*) ratio vs. the NoS for I-shaped pillar consideration (g_k_fablim_ = 2 µm, w_k_fablim_ = 6 µm), where *L*($\gamma_{k}$) is the minimum length using the ideal gamma per section, while *L*(*γ’*) is the minimum length using the same gamma across all sections.

**Table 1 micromachines-15-00405-t001:** Definitions of important variables for the design of multi-section DLD devices.

Variables	Definition
*D_ck_*	Critical diameter of the particle
*D_p_max_*	Biggest particle that will be separated
*D_p_min_*	Smallest particle that will be separated
*w*	Diameter of the pillar
*g*	Gap between the pillar (in the lateral direction)
*D_x_*	Center-to-center distance in the flow direction
*D_y_*	Center-to-center distance in the lateral direction
*L*	Total length of the device
*NoS*	Number of sections
*SSF*	Section-scaling factor
*N*	Number of rows required for one column shift
*γ*	Pillar diameter to gap ratio (*γ = w/g*)
*β*	1.1—design tolerance
*θ*	Gradient angle (tan(*θ*) = 1/N)
*m*	1 (Number of columns to be displaced)

**Table 2 micromachines-15-00405-t002:** Comparison for different fabrication limit cases.

Cases		DLD Device Length
**Case 1:** **I-shaped pillar;** ** $g_{k\_fablim}=2\;µm;\;w_{k\_fablim}=$ ** **6 µm;**	*D_p_min_* = 0.01 µm; *D_p_max_* = 10 µm	*L* = 60 m Dynamic range = 1000
**Case 2:** Circle-shaped pillar; $g_{k\_fablim}$ = $w_{k\_fablim}=$ 0.1 µm;	*D_p_min_* = 1 µm; *D_p_max_* = 10 µm;	*L* ~ 0.3 mm Dynamic range = 10
	*D_p_min_* = 0.01 µm; *D_p_max_* = 10 µm;	*L* ~ 41 mm Dynamic range = 1000
**Case 3:** Circle-shaped pillar; $g_{k\_fablim}$ = $w_{k\_fablim}=$ 10 µm;	*D_p_min_* = 1 µm; *D_p_max_* = 100 µm;	*L* ~ 8 mm Dynamic range = 100

## Data Availability

Data is contained within the article (and Appendix A).

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
