# Peer review of "Geometry Scaling for Externally Balanced Cascade Deterministic Lateral Displacement Microfluidic Separation of Multi-Size Particles†"

_micromachines, 2024, doi:10.3390/mi15030405_

Round 1

Reviewer 1 Report

Comments and Suggestions for Authors

The authors constructed a model to guide the design of DLD-based microfluidic device. Several factors have been discussed. This is an interesting topic in microfluidics. Here are some comments:

1. The introduction should be more focused. The authors are suggested to provide application examples of DLD devices in real scenarios to demonstrate their importance.

2. How should the values of Wk_fablim in the model be considered and set to represent different shapes of pillars, such as circular, triangle and square shapes?

3. Experimental results are required in Section 7 “case study” to verity the design of DLD device using the model.

Author Response

Response: We thank the reviewer for their insightful comments and amazingly helpful editing notes. We have addressed all of the comments with the discussion below and changes to the manuscript marked in blue text.

Reviewers' Comments to the Authors:

Reviewer: 1
Comments to the Author
The authors constructed a model to guide the design of DLD-based microfluidic device. Several factors have been discussed. This is an interesting topic in microfluidics.

Here are some comments:

  1. The introduction should be more focused. The authors suggested providing application examples of DLD devices in real scenarios to demonstrate their importance.

The introduction has been heavily edited to be more focused, and applications requiring DLD devices to perform size fractionation have been expanded to highlight the importance of DLD technology.

  1.     How should the values of Wk_fablim in the model be considered and set to represent different shapes of pillars, such as circular, triangle and square shapes?

To help the reader better understand wk_fablim, we have expanded the description in section 6.1, paragraph 1, to make this clearer. To clarify this, we have defined the fabrication resolution as FR, and defined wk_fablim as a function of FR for different pillar shapes.

  1.     Experimental results are required in Section 7 “case study” to verity the design of DLD device using the model.

Like all sections of this model-based paper, section 7 is based on detailed simulations, with the results summarized in Table 2. However, in the prior version of this manuscript, Table 2 was misaligned with Section 7 due to an error in translating the manuscript to the journal template. This alignment has been corrected and the simulation results in Table 2 should now appear within section 7. While we appreciate that some readers may wish to see experimental results from fabricated devices, we believe that experimental analysis of each case study would require a separate paper for each case and is beyond the scope of this manuscript.

Reviewer 2 Report

Comments and Suggestions for Authors

This paper presented only the modeling results. It's relevance will be seen if supported by experimental works also.

In Figure 10 b and e, the variation of L with Nos is expected to vary smooth as the results are only theoretical in nature, as seen in other graphs. An explanation in this regard will be necessary.

Author Response

Response: We thank the reviewer for the insightful comments and amazingly helpful editing notes. We have addressed all of the comments with the discussion below and changes to the manuscript marked in blue text.

Reviewer: 2
Comments to the Author

1) This paper presented only the modeling results. It's relevance will be seen if supported by experimental works also.

We acknowledge this paper presents a model for a new design approach as well as extensive simulation results from this model. We agree that the impact of this work will be magnified once this paper is made available to the research community who can expand on this work with experimental design and characterization.

We have modified the Introduction (Paragraph 3) and Conclusion sections to better clarify the scope of this work and the impact it can have on future experimental design efforts.

2) In Figure 10 b and e, the variation of L with Nos is expected to vary smooth as the results are only theoretical in nature, as seen in other graphs. An explanation in this regard will be necessary.

The results in Fig 10 show cases where the ɣ parameter is optimized individually for each section. These are not the results of a single calculation evaluated over NoS, but rather each section is discretely optimized. Because the value of ɣ can change for each section, the overall plots vs. NoS may not smoothly match the trendline (the less optimal ɣ changes with each section, the smoother the plot).

We have added more explanation to section 6.2 to better clarify this.

Reviewer 3 Report

Comments and Suggestions for Authors

This manuscript outlines a design strategy aimed at enhancing the dynamic range of particle separation in externally balanced cascade deterministic lateral displacement (DLD) devices. A mathematical model is introduced to systematically address design tradeoffs and practical limitations inherent in  microfabrication. Utilizing their proposed model, the authors conducted an analysis demonstrating that cascade DLD devices can achieve both high resolution and a wide dynamic range of particle separation.  

The reviewer judges this work suitable for publication in Micromachines if the authors address the following concerns:

(1)

The reviewer suggests eliminating the phrase “for IOT Environmental Sensing” from the title, asserting that the significance of the concept extends beyond the realm of IOT environmental sensing.

(2)

The meaning of “externally balanced” in the title remains unclear. Though Figure2’s caption hints at a connection to hydraulic resistance, the reviewer recommends clarifying this term earlier in the Introduction section or elsewhere.

(3)

The reviewer questions the novelty of the work, particularly in Section 3.1, where the authors claim to differentiate their approach by varying pillar geometry parameters (i.e., pillar gap and pillar size). However, the reviewer believes that cascading multiple sections with decreasing pillar gap has been previously reported in literature. For example, see the following paper: RSC Adv. 2017, 7, 35516.

(4)

The References section requires improvement. Several duplications are noted, such as refs. 26, 28, and 36 being the same. Additionally, refs. 20 and 21, as well as 23, 29, and 45, should be consolidated. The journal name is missing for ref. 42, and there are discrepancies in the journal name for ref. 26. These issues need to be rectified for a more accurate and polished reference section.

Author Response

Response: We thank the reviewer for the insightful comments and amazingly helpful editing notes. We have addressed all of the comments with the discussion below and changes to the manuscript marked in blue text.

Reviewer: 3
Comments to the Author

This manuscript outlines a design strategy aimed at enhancing the dynamic range of particle separation in externally balanced cascade deterministic lateral displacement (DLD) devices. A mathematical model is introduced to systematically address design tradeoffs and practical limitations inherent in  microfabrication. Utilizing their proposed model, the authors conducted an analysis demonstrating that cascade DLD devices can achieve both high resolution and a wide dynamic range of particle separation. 

The reviewer judges this work suitable for publication in Micromachines if the authors address the following concerns:

(1) The reviewer suggests eliminating the phrase “for IOT Environmental Sensing” from the title, asserting that the significance of the concept extends beyond the realm of IOT environmental sensing.

We agree that multi-size particle separation with DLD technology has broad applications beyond the realm of IOT environmental sensing. We have modified the title as suggested.

(2) The meaning of “externally balanced” in the title remains unclear. Though Figure2’s caption hints at a connection to hydraulic resistance, the reviewer recommends clarifying this term earlier in the Introduction section or elsewhere.

We thank the reviewer for identifying this. We have modified the introduction to better clarify this term. Moreover, this term is further described in section 3 with methods for achieving external balance discussed in the last paragraph of section 3 (also highlighted as blue).

(3) The reviewer questions the novelty of the work, particularly in Section 3.1, where the authors claim to differentiate their approach by varying pillar geometry parameters (i.e., pillar gap and pillar size). However, the reviewer believes that cascading multiple sections with decreasing pillar gap has been previously reported in literature. For example, see the following paper: RSC Adv. 2017, 7, 35516.

We really appreciate this issue being raised because it made us aware that our manuscript did not fully explain internally and externally balanced DLD devices, nor did we adequately explain the differences between them, which is key to understanding the novelty of this work. The paper “RSC Adv. 2017, 7, 35516” is a good example of an internally balanced device, which is a term we’ve introduced to contrast with our externally balanced device presented in this paper. Internally balanced DLD devices contain different cascased sections, each with different pillar angle or gap or size modifying the critical size (separation size) within each section, as the reviewer correctly noted. The main limitation of any internally balanced device is that it must continue to process the large particles separated in a prior section. As a result, the pillar gap can never be set smaller than the largest particle to avoid clogging the device. And, without reducing the pillar gap, devices will become very long (potentially meters in length, as covered within this manuscript) in order to separate smaller and smaller particles. Thus, the main limitation of any existing multi-size separation DLD device (those we define as internally-balanced) is they can only separate particles over a small size range. In contrast, the externally-balanced approach presented in this manuscript, removes the larger particles separated by each section, and this allows the pillar gap of the next section to be scaled down to separate smaller particles. Thus, as the results of this manuscript show, our externally-balanced method uniquely allows separating particles having a large range of sizes, or as we say in the manuscript, we greatly expand the dynamic range for size separation. That is the novelty of this work, and that is something that no internally balanced device can achieve.

To clarify this within our manuscript, we have made significant modification to the Introduction and greatly expanded section 3.1 to clarify the novelty of this work by describing internally-cascaded multi-size DLD devices (like the paper cited in this review comment, which we now reference in the Introduction) and contrasting them to the capabilities of our externally balanced, cascaded section, DLD design.

(4) The References section requires improvement. Several duplications are noted, such as refs. 26, 28, and 36 being the same. Additionally, refs. 20 and 21, as well as 23, 29, and 45, should be consolidated. The journal name is missing for ref. 42, and there are discrepancies in the journal name for ref. 26. These issues need to be rectified for a more accurate and polished reference section.

We apologize for not catching the errors in references that occurred while transferring this manuscript to the journal’s Word template. These issues have all been fixed in the revised manuscript.

Round 2

Reviewer 1 Report

Comments and Suggestions for Authors

The authors have addressed most of my comments and the revised manuscript can be accepted for publication.